# Alleviation of Adverse Effects of Drought Stress on Growth and Nitrogen Metabolism in Mungbean (*Vigna radiata*) by Sulphur and Nitric Oxide Involves Up-Regulation of Antioxidant and Osmolyte Metabolism and Gene Expression

**DOI:** 10.3390/plants12173082

**Published:** 2023-08-28

**Authors:** Huida Lian, Cheng Qin, Jie Shen, Mohammad Abass Ahanger

**Affiliations:** 1Department of Life Sciences, University of Changzhi, Changzhi 046000, China; lianhuidada@163.com (H.L.); qc332084910@163.com (C.Q.);; 2College of Life Science, Northwest A&F University, Yangling 712100, China

**Keywords:** oxidative damage, nitrogen metabolism, antioxidants, secondary metabolites, gene expression, drought, nitric oxide, sulphur

## Abstract

The influence of drought induced by polyethylene glycol (PEG) and the alleviatory effect of nitric oxide (50 µM) and sulphur (S, 1 mM K_2_SO_4_) were studied in *Vigna radiata*. Drought stress reduced plant height, dry weight, total chlorophylls, carotenoids and the content of nitrogen, phosphorous, potassium and sulphur. The foliar applications of NO and sulphur each individually alleviated the decline, with a greater alleviation observed in seedlings treated with both NO and sulphur. The reduction in intermediates of chlorophyll synthesis pathways and photosynthesis were alleviated by NO and sulphur. Oxidative stress was evident through the increased hydrogen peroxide, superoxide and activity of lipoxygenase and protease which were significantly assuaged by NO, sulphur and NO + sulphur treatments. A reduction in the activity of nitrate reductase, glutamine synthetase and glutamate synthase was mitigated due to the application of NO and the supplementation of sulphur. The endogenous concentration of NO and hydrogen sulphide (HS) was increased due to PEG; however, the PEG-induced increase in NO and HS was lowered due to NO and sulphur. Furthermore, NO and sulphur treatments to PEG-stressed seedlings further enhanced the functioning of the antioxidant system, osmolytes and secondary metabolite accumulation. Activities of γ-glutamyl kinase and phenylalanine ammonia lyase were up-regulated due to NO and S treatments. The treatment of NO and S regulated the expression of the *Cu/ZnSOD*, *POD*, *CAT, RLP*, *HSP70* and *LEA* genes significantly under normal and drought stress. The present study advocates for the beneficial use of NO and sulphur in the mitigation of drought-induced alterations in the metabolism of *Vigna radiata*.

## 1. Introduction

Drought stress is a global problem and one of the key threats to global food security. It adversely affects the growth, development and distribution of plants [1]. Drought results in significant declines in root growth, photosynthesis, transpiration, nutrient uptake and assimilation, enzyme activity and yield productivity [1,2,3]. Drought triggers symptoms including leaf senescence, drooping and rolling, etiolation, wilting, premature flower fall and yellowing of leaves [4]. Drought is characterised by significant declines in tissue water potential, turgor, stomatal movements and cellular proliferation and enlargement [5]. In addition to this, drought stress triggers the excess generation of reactive oxygen species (ROS) and methylglyoxal (MG), thereby resulting in oxidative damage to key macromolecules including proteins, amino acids and nucleic acids [1]. This oxidative stress-induced modulation can be damaging to plant growth and development, reflected in significant alterations in physiological and biochemical attributes [1]. These ROS- and MG-induced alterations are alleviated by up-regulating the tolerance mechanisms including the antioxidant system, osmolyte accumulation and glyoxylase system [6]. It has been reported that the up-regulation of tolerance mechanisms results in an enhanced potential to withstand the damaging effects of drought stress [7,8,9].

Sulphur (S) is an important mineral element required for the growth and development of plants, as well as tolerance to stresses. Sulphur forms a key constituent of amino acids like cysteine and methionine; vitamins like biotin and thiamine; and other compounds like glutathione, lipoic acid, glucosinolates, etc. [10]. It has been reported that sulphur modulates the tolerance mechanisms to alleviate the damaging effects of stresses on key metabolic pathways, and hence, plant performance is the least affected [11]. Stresses affect the uptake, transport and assimilation of sulphur significantly, thereby imposing a significant damaging influence on key pathways like photosynthesis and stress-tolerant mechanisms [12]. The supplementation of sulphur has been reported to strengthen the tolerance mechanisms in different plants individually as well as interactively with other molecules [13].

Nitric oxide (NO) is an important redox-active signalling molecule regulating germination, enzyme functioning, mineral uptake and assimilation, photosynthesis, osmolyte accumulation, gene expression and stress tolerance [14,15,16]. The production of NO in plants occurs through the nitric oxide synthase (NOS) pathway, the nitrate reductase pathway and other enzymatic and non-enzymatic pathways [14,17,18]. Stresses trigger the generation of NO, and the exogenous treatment of NO has been reported to optimise the endogenous NO for better stress tolerance [19,20], aside from the maintenance of NO concentration through the supplementation of mineral elements and phytohormones [19,21]. It has been reported that the treatment of NO through foliar applications or priming potentiates the tolerance mechanism to alleviate the damaging effects of stresses in different plants [15,22]. The application of NO alleviated the damaging effects of heavy metal [15], salinity [19], heat stress [20] and drought [23] by up-regulating the antioxidant system, thereby preventing damage to key macromolecules and their functioning. Transcriptomic studies of sulphur and NO-treated plants have revealed significant modulation in the genes regulating key metabolic pathways, including the tolerance mechanisms [24,25]

*Vigna radiata,* commonly known as mung bean, is an important edible legume crop grown throughout the world for seeds and has a long history as a traditional medicine. It is rich in proteins, minerals, vitamins, dietary fibre and bioactive compounds, including polyphenols, peptides, etc. Mungbean has been used in the treatment of several disorders including hyperglycaemia, hypertension, cancer, melanogenesis, etc. [26]. Drought reduces its growth and productivity significantly. The present study was aimed at investigating the beneficial role of sulphur and NO in alleviating the damaging effects of drought stress by assessing the modulation of tolerance mechanisms and gene expression patterns. Modulations in tolerance mechanisms, oxidative stress parameters, photosynthesis and enzyme functioning due to S and NO were evaluated.

## 2. Results

Results showing the effects of NO and sulphur under drought stress on the δ-ALA, GSA, total chlorophylls, carotenoids and photosynthesis are shown in Figure 1. Relative to the control, drought stress reduced δ-ALA by 37.12%, GSA by 47.67%, total chlorophylls by 40.29%, carotenoids by 41.91% and photosynthesis by 39.35%. Both NO and S alleviated this decline to significant levels, with S being much more effective. The maximal alleviation of the drought-induced decline was observed in plants treated with both NO and sulphur. Relative to the control, the decline in δ-ALA, GSA, total chlorophylls, carotenoids, photosynthesis and Fv/Fm was only 7.78%, 5.91%, 11.44%, 12.20% and 13.66%, respectively, in PEG + NO + sulphur-treated plants (Figure 1 and Figure 2A–E).

Drought stress resulted in significant increases in H_2_O_2_ (101.37%), O_2_^−^ (47.10%) and lipid peroxidation (139.04%) compared to the control. The foliar application of NO and the supplementation of sulphur through roots declined the levels of H_2_O_2_, O_2_^−^ and lipid peroxidation, with maximal reduction observed in plants treated with both NO and sulphur. Relative to drought stress plants, a decline of 73.25% in H_2_O_2_, 71.21% in O_2_^−^ and 80.81% in lipid peroxidation was observed in D + sulphur + NO-treated plants (Figure 3A–C).

The activity of protease and lipoxygenase was reduced due to the treatment of NO and sulphur compared to the control plants. Drought stress resulted in a significant increase in the activity of protease (79.24%) and lipoxygenase (148.93%) compared to the control. The application of NO and sulphur to drought-stressed plants reduced the activity of protease and lipoxygenase by 36.04% and 46.37%, respectively, compared to the drought-stressed plants (Figure 4A,B).

The content of proline, sugars and glycine betaine increased by 49.53%, 35.92% and 33.77%, respectively, due to drought compared to the control plants. The treatment using NO and sulphur for drought-stressed plants further increased the content of proline, sugar and glycine betaine and the activity of γ-glutamyl kinase. Relative to the control, proline, sugars, glycine betaine and activity of γ-glutamyl kinase showed an increase of 78.82%, 68.93%, 59.45% and 47.13% in D + NO, 103.31%, 83.98%, 79.97% and 59.08% in D + sulphur and 141.25%, 105.33%, 97.06% and 74.90% in D + NO + sulphur-treated plants (Figure 5A–D).

The activity of NR, GS and GOGAT was reduced by 31.45%, 48.97% and 48.68% due to drought; however, the application of NO and the supplementation of sulphur individually, as well as combinedly, alleviated the decline. The combined application of NO and sulphur maximally alleviated the decline in NR, GS and GOGAT by 27.32%, 67.10% and 62.53%, respectively, compared to the drought-stressed plants (Figure 6A–C).

The activity of SOD, CAT, APX, DHAR and GR increased by 63.39%, 23.35%, 19.83%, 38.89% and 34.41%, respectively, due to drought stress and the application of NO and sulphur to drought-stressed plants further enhanced the activities of these enzymes. Relative to the control, maximal increases in the activities of SOD, CAT, APX, DHAR and GR were158.03%, 97.81%, 119.28%, 94.52% and 78.26%, respectively, in D + NO + sulphur-treated plants (Figure 6). The content of AsA, GSH and tocopherol increased by 23.09%, 21.32% and 30.00%, respectively, due to drought stress compared to the control. The foliar application of NO and the supplementation of sulphur further increased the content of AsA, GSH and tocopherol attaining maximal increases of 71.47%, 86.20% and 95.00% in D + S + NO-treated plants (Figure 7).

Drought stress resulted in increases of 122.36% and 191.63% in hydrogen sulphide and nitric oxide compared to the control. The treatment using NO, sulphur and NO + sulphur for drought-stressed plants increased the hydrogen sulphide and nitric oxide content compared to the control. Relative to the control, hydrogen sulphide and nitric oxide increased by 65.46% and 103.08% in D + sulphur+ NO-treated plants (Figure 8).

Drought increased phenols (14.01%), flavonoids (13.93%) and the activity of PAL (26.00%) compared to the control. The treatment with NO and sulphur significantly enhanced the phenols, flavonoids and the activity of PAL. The maximal enhancement of 50.12% in total phenols, 45.45% in flavonoids and 90.66% in the activity of PAL was observed in D + sulphur+ NO-treated plants compared to the control (Figure 9).

Drought reduced the content of N, P, K and S by 47.26%, 42.63%, 45.67% and 40.91%, respectively, compared to the control; however, the application of NO and sulphur alleviated the decline to significant levels. Compared to drought-stressed plants, maximal alleviations of 55.19%, 47.35%, 66.83% and 38.72% were observed in N, P, K and S, respectively (Figure 10).

The expression of *VrCu-Zn/SOD*, *VrPOD* and *VrCAT* in the leaves of drought-stressed plants increased by 1.69-, 4.15- and 7.35-fold, respectively, compared to the control. The treatment with NO and sulphur significantly enhanced the gene expression of *VrCu-Zn/SOD*, *VrPOD* and *VrCAT*. The maximal enhancement in the expression of *VrCu-Zn/SOD*, *VrPOD* and *VrCAT* was observed in D + sulphur + NO-treated plants compared to the control (Figure 10). Drought stress markedly reduced the expression of *VrRLP* genes compared to the control; however, NO and sulphur treatment significantly enhanced gene expression of *VrRLP.* Maximal enhancement by3.62-fold in *VrRLP* was observed in D + sulphur+ NO-treated plants compared to the control. Drought stress resulted in an increase of 1.83- and 2.67-fold in the expression of *VrHSP70* and *VrLEA* compared to the control. The treatment with NO, sulphur and NO + S for drought-stressed plants decreased the expression of *VrHSP70* and *VrLEA* compared to drought. Relative to drought, the expression of *VrHSP70* and *VrLEA* decreased by 66.53% and 60.95% in D + sulphur+ NO-treated plants (Figure 11).

The correlation analysis showed highly negative correlations among H_2_O_2_, O_2_^•−^and proline, SOD, GB, Chl and Car. The content of H_2_O_2_ was negatively correlated with SOD (*p* < 0.05) and O_2_^•−^ was negatively correlated with proline (*p* < 0.05). MDA was significantly negatively correlated with flavonoids (*p* < 0.05). In contrast, significant positive relationships were found among NO, N and GR. Overall, photosynthetic and other physiological parameters of mungbean leaves are regulated by sulphur and NO under drought stress (Figure 12).

## 3. Discussion

Drought is a global problem and immensely affects global food security. Rapid climate change has aggravated the situation, and in the near future, it is expected to worsen. Therefore, management strategies have been devised to tackle the drought-induced growth changes in plants. In the present study, the role of the exogenous application of NO and the supplementation of sulphur was individually and combinedly evaluated under drought stress in *Vigna radiata* (Figure 1). The foliar application of NO and the supplementation of sulphur enhanced plant growth with a more obvious effect demonstrated in plants treated by their combined treatments. Drought reduced the content of GSA, δ-ALA, total chlorophyll and carotenoids significantly; however, it was observed that NO and S treatments alleviated the decline by a considerable extent, with maximal alleviation observed due to their combined treatment. Individually, S proved much more effective than NO. An earlier drought stress-induced decline in GSA, δ-ALA, total chlorophyll and carotenoids has been reported in rice [27], pea [28] and tomato [3]. Stress-exposed plants exhibiting a decline in the intermediates of the chlorophyll biosynthesis pathway and carotenoids show a significant decline in photosynthesis and water-use efficiency [29]. It has been reported that stresses down-regulate the activity of enzymes and the expression of genes involved in the chlorophyll synthesis pathway [27,29,30]. The reduced chlorophyll synthesis was under stress, including drought results from the up-regulation of chlorophyllase [28]. In the past, the alleviation of the decline in chlorophyll and carotenoid synthesis due to NO [31] and S [32] under drought stress has been reported. However, the alleviatory effect of the combined treatment of NO and sulphur against a drought-induced decline in chlorophyll synthesis and photosynthesis has not been reported. In salt-stressed *Brassicajuncea*, the combined treatment of NO and sulphur has been reported to enhance chlorophyll synthesis, Rubisco activity and photosynthesis [19]. In the present study, the alleviation of a drought-induced decline in total chlorophylls, carotenoids, photosynthesis and PSII activity due to NO and sulphur may be attributed to a reduction in the ROS accumulation, improved water content, increased Rubisco synthesis and mineral assimilation. A drought-induced reduction in water potential and D1 protein levels significantly contributes to declined chlorophyll and photosynthetic functioning [33].

Reduced growth due to PEG-induced drought stress was related to the excessive generation of ROS, including H_2_O_2_ and O_2_^−^, resulting in oxidative damage to key macromolecules including proteins and lipids. In addition, the oxidative effects of drought were also evident in terms of increased lipid peroxidation and the activity of protease and lipoxygenase. Drought-induced enhancement of the ROS has previously been reported by Antonic et al. [34] in *Impatiens walleriana* and Hajihashemi and Sofo [35] in *Stevia rebaudiana*, resulting in a significant decline in growth. Recently, in PEG-stressed *Cucumis melo* genotypes, Mehmandar et al. [36] have demonstrated a significant enhancement in the H_2_O_2_ and lipid peroxidation resulting in declined growth, biomass and yield production. Excess ROS affects membrane stability, electron transport, photosynthesis, protein cross-linking, protein synthesis, ion transport and enzyme activity [37]. Exogenously applied NO and S supplementation alleviated the oxidative effects of PEG by reducing the H_2_O_2_, O_2_^−^ and lipid peroxidation to significant levels. Excess ROS in chloroplasts affects the PSII assembly, thereby significantly influencing the photosynthetic functioning; however, the NO- and sulphur-mediated decline in ROS reflects their beneficial role in the prevention of damage to the photosynthetic apparatus. The reduced generation of ROS and lipid peroxidation due to NO [23] and sulphur [38] has been reported earlier. However, the influence of their combined application has not been reported. Plants maintaining lower ROS levels following NO and sulphur treatments have been reported to exhibit increased photosynthetic functioning, and as a result, increased water-use efficiency [22,38,39,40]. In addition, NO- and sulphur-treated plants exhibited a significant decline in the activity of lipoxygenase and protease, which were up-regulated due to drought stress. An earlier increase in the activities of protease and lipoxygenase due to drought stress has been reported by Ahanger et al. [3] and Shreya et al. [41]. The increased activity of protease and lipoxygenase reflects the increased damage to proteins and lipids, thereby causing animbalance in key cellular structures and their functioning [42,43]. However, a NO- and sulphur-mediated decline in the activities of protease and lipoxygenase confirms their beneficial impact in protecting the sensitive molecules in plants. Lipoxygenase mediated oxidation-generated fatty acid hydroperoxides from the polyunsaturated fatty acids [43] and proteases degrade proteins that have been denatured, damaged and aggregated due to stress [40]. Reduced activity of lipoxygenase due to exogenous treatment of NO has been reported in *Hordeum vulgare* [44]. Reports showing the effects of S and NO + sulphur supplementation on protease and lipoxygenase activity under drought stress are not available.

The damaging effects of excessive ROS are generated due to drought-stressed plants up-regulating the antioxidant system. In the present study, the activity of antioxidant enzymes, including SOD, CAT, APX, DHAR and GR, was significantly increased due to PEG-induced drought stress. The up-regulation of the activity of antioxidant enzymes due to PEG-induced drought stress has been reported in different crop plants including *Triticum aestivum* L [7], soyabean [45], *Stevia rebaudiana* [22] and *Arachis hypogea* [39]. Recently, it has been reported that the PEG-induced drought significantly up-regulates the activity of antioxidant enzymes, and plant genotypes exhibiting increased activities better tolerate the adverse growth conditions [36]. Increased functioning of antioxidant enzymes results in quick scavenging of toxic radicals, which reflects the protection of membranes, enzymes, proteins, lipids, nucleic acids, etc. [46,47,48]. In the present study, the exogenous application of NO and the supplementation of S to PEG-stressed seedlings significantly up-regulated the activities of antioxidant enzymes assayed, thereby strengthening the potential to neutralise ROS. This leads to greater protection of major cellular structures and their functioning. In corroboration of our results, the findings of Rezayian et al. [49] and Usmani et al. [50] have also demonstrated that NO and S application, respectively, up-regulates the antioxidant enzyme activity under drought stress. However, the effect of combined NO and S treatments on the antioxidant system under drought has not worked. Among the antioxidant system, the enzymes SOD specifically neutralise the superoxide while H_2_O_2_ is eliminated by the efficient functioning of CAT in cytosol or ascorbate-glutathione cycle in mitochondria and chloroplast [48].The ascorbate-glutathione cycle functioning depends on the APX, DHAR, GR, AsA and GSH components. In the present study, AsA and GSH were increased significantly due to NO and sulphur treatments, thereby contributing to photosynthetic protection by maintaining low concentrations of ROS, NADP/NADPH ratio and redox homeostasis. Ascorbate, glutathione and tocopherol scavenge radicals, thereby protecting plants from oxidative effects. Increased levels of AsA [51], GSH [52] and tocopherol [53] due to drought have been reported earlier, thereby assisting in neutralising the toxic radicals and protecting the major plant functions. The alteration of the functioning of biosynthesis and the degradation-related genes result in the accumulation of tocopherol [54].

Drought induced by PEG resulted in are duction in the activities of the enzymes of nitrogen assimilation including NR, GS and GOGAT. Drought potentially reduced the activities of NR, GS and GOGAT in *Triticum aestivum* [55] and *Brassica juncea* [56]. Increased uptake and the subsequent metabolism of N contribute to growth improvement and stress tolerance by improving the synthesis of metabolites and amino acids [57]. Increased N uptake and metabolism significantly contributes to drought tolerance through enhanced osmolyte synthesis and antioxidant functioning [58], resulting in improved photosynthesis [59]. Similar to our results, the alleviation of PEG that triggered a decline in the activities of NR, GS and GOGAT due to the exogenous application of NO has been shown in alfalfa [60]. The sulphur-mediated enhancement of the NR activity is strongly correlated with yield [61]. A deficiency in S adversely affects the N assimilation, and hence affects the N use efficiency [62]. The combined treatments of NO and sulphur maximally enhanced the activity of the N assimilation pathway enzymes.

Drought stress resulted in an increase in the phenolics and flavonoids, and similar to these findings, earlier published reports are also available [63,64,65]. The accumulation of phenolics and flavonoids effect the antioxidant potential of plants, thereby contributing to stress mitigation and the regulation of growth under adverse conditions [7,64]. Phenylalanine ammonia-lyase is one of the enzymes essential for allocating enough carbon from phenylalanine for the biosynthesis of secondary metabolites [66,67], and NO- and sulphur-induced increases in PAL activity contributed considerably to phenol and flavonoid synthesis under PEG stress. An increase in secondary compounds due to NO treatments under drought has been observed in *Viciafaba* [68] and soybean [69]. Treatments of sulphur increase the content of secondary metabolites in plants under water-deficit conditions [70]; however, reports discussing the influence of combined NO and sulphur treatments are not available. The improved functioning of an antioxidant system due to the supplementation of S and the foliar application of NO was also evident in the expression patterns of SOD, CAT and POD. Similar to our results, Danyali et al. [71] and Rahimi et al. [72] have also observed the up-regulation of the expression of antioxidant genes, including *Cu/ZnSOD*, *CAT* and *POD* under drought stress. The increased expression of antioxidant genes in NO- and sulphur-treated plants justifies their beneficial interactive role in drought stress alleviation. An increase in the expression of antioxidant genes due to NO [73,74] and sulphur [38] has been reported in several crops under different stresses. However, the combined effect under drought has not been evaluated. In addition, the expression of LEA [75] and *HSP70* significantly increased due to drought exposure, while S and NO treatments imparted a decline under drought. *LEA* is a drought-responsive gene and its overexpression improves drought tolerance [76], and it has a key role in protecting cells from adverse effects of stresses [77]. Drought triggers the expression of *HSP70* [78], and the overexpression of *HSP70* has been demonstrated to protect plants from the ill effects of drought stress [79]. Moreover, drought reduced the expression of RubisCO-like protein (RLP), while sulphur and NO treatments resulted in increased activity of RLP, reflected in the increased photosynthetic assimilation.

Plants treated with NO and sulphur exhibited an increase in proline, sugars and glycine betaine. Research reports corroborating our results of increased proline, sugar and glycine betaine under drought are available [80,81,82]. The osmolytes help plants to maintain tissue water potential, protect enzyme degradation and functioning, prevent photosynthetic inhibition and scavenge ROS [83,84,85]. Exogenous NO resulted in an increased proline accumulation in barley causing an increase in photosynthesis and fluorescence parameters [86]. Exogenous NO modulates the functioning of glycine betaine biosynthesis pathways, resulting in its increased accumulation and improved drought tolerance [87]. The increased accumulation of proline by NO and sulphur application may have resulted due to the significant enhancement in the activity of γ-glutamyl kinase activity. Earlier plants exhibiting the enhanced activity of γ-glutamyl kinase also accumulated increased proline [88].

The endogenous concentration of NO and HS showed significant increases due to drought stress. It was interesting to note that the exogenous NO application and sulphur supplementation reduced both NO and HS compared to the drought-stressed plants; however, they were higher than the control. Both NO and HS act as signalling molecules and share crosstalk for the regulation of plant processes, including germination, senescence, ripening and stress alleviation [89,90]. Several reports are available showing increased NO [86] and HS [87] under drought stress conditions. The treatment of NO increases endogenous HS, and maintaining optimal levels of NO and HS is essential for stress signalling and the activation of tolerance mechanisms [91,92]. Sulphur increased endogenous HS in *Brassica juncea* by up-regulating the functioning of biosynthesising enzymes [93]. The treatment of NO improves sulphur-use efficiency and the photosynthetic functioning of rice under heat stress [20]. Our study showed that exogenous NO and sulphur enhanced endogenous NO and HS more due to their combined treatments. It is often the case that both NO and HS share a crosstalk mechanism that regulates their endogenous levels and also influences the tolerance mechanisms; however, further studies are required.

## 4. Material and Methods

Seeds of mungbean (*Vigna radiata* cultivar Jin 8) were surface-sterilized using 0.001% HgCl_2_ for 5 min followed by washing five times with distilled water. Sterilized seeds were sown in earthen pots. The pots (22 cm diameter) were filled with sand and compost (3:1), and at the time of sowing, the pots were wetted with 300 mL full-strength Hoagland solution. After germination, thinning was carried out and three healthy and uniformly growing seedlings were maintained in each pot. The composition of Hoagland solution was 1. 0 mM NH_4_NO_3_, 0.4 mM KH_2_PO_4_, 1.0 mM K_2_SO_4_, 0.5 mM K_2_HPO_4_, 3.0 mM CaCl_2_, 0.5 mM MgSO_4_, 0.2 mM Fe-NaEDTA, 14 µM H_3_BO_3_, 5.0 µM MnSO_4_H_2_O, 3.0 µM ZnSO_4_.7H_2_O, 0.7 µM CuSO_4_.5H_2_O, 0.7 µM (NH_4_)_6_Mo_7_O_24_ and 0.1 µM CoCl_2_. Ten days after seedling growth, drought stress was induced by 20% polyethylene-6000 (PEG-6000) dissolved in the Hoagland nutrient solution, and the foliar application of nitric oxide (50 µM NO in the form of sodium nitroprusside) was also started after ten days. Pots receiving the Hoagland solution without K_2_SO_4_, PEG and NO served as the control. The Hoagland solution was added to each pot on alternate days. The foliar application of NO was also carried out on every alternate day. Hence, the overall treatments in the present experiment included: (a) control, (b) drought (D; PEG (20%), (c) D + S (1 mM K_2_SO_4_), (d) D + NO and (e) D + S + NO.

### 4.1. Estimation of Total Chlorophylls, Carotenoids and Net Photosynthesis

For the extraction of photosynthetic pigments, 100 mg of fresh leaf tissue was homogenised in 80% acetone. After centrifuging, the extract optical density of the supernatant was taken at 480, 645 and 663 nm [94]. For the measurement of net photosynthesis (*Pn*), the LI-6400 photosynthesis system (Li-Cor, Lincoln, NE, USA) was used and measurements were carried between 09:00–12:00 h.

### 4.2. Estimation of Glutamate 1-Semialdehyde (GSA) and δ-Amino Levulinic Acid (δ-ALA)

The content of GSA was estimated according to Kannangara and Schouboe [95]. Fresh (200 mg) two sets of tissue were taken, in which one set was extracted in 0.1 N HCl while another one was incubated in 500 µM gabaculine in 100 mM 2-(4-morpholino)ethanesulfonic acid (MES; pH 7.0) for hours under light. After extraction, homogenate was centrifuged for 10 min at 15,000× *g* at 4 °C. The supernatant was mixed with HCl and 3-methyl-2-benzothiazolinonehydrazone (MBTH), followed by incubation in a boiling water bath for 2 min. After cooling the samples, distilled water and FeCl_3_ were added, and the mixture was thoroughly mixed. The optical density was recorded at 620 nm.

The method described by Harel and Klein [96] was followed for the estimation of δ-ALA content. Briefly, two sets of fresh leaf samples (200 mg) were taken, in which one set was incubated for 4 h in 60 mM levulinic acid (LA) under light while another was immediately extracted in 5 mL 1 M sodium acetate buffer (pH 4.6). Homogenate was centrifuged for 10 min at 15,000× *g*, and to 1 mL supernatant, distilled water and acetyl-acetone were added. The mixture was kept in a boiling water bath for 10 min. After cooling the samples at room temperature, Ehrlich’s reagent was added, and samples were vortexed and optical density was read at 555 nm. ALA synthesized during a 4 h incubation period was measured by subtracting the 0 h ALA from a 4 h ALA content.

### 4.3. Estimation of Proline and Glycine Betaine, Free Sugars and Activity of γ-Glutamyl Kinase Activity

For the estimation of proline content method of Bates et al. [97] was used and dry powdered tissue was extracted in 3% sulphosalicylic acid. Toluene was used to separate proline and the absorbance was taken at 520 nm. Glycine betaine (GB) and free sugars were estimated according to Grieve and Grattan [98] and Jain and Guruprasad [99], respectively. The activity of γ-glutamyl kinase (γ-GK, EC 2.7.2.11) was assayed by extracting the fresh 500 mg tissue in a Tris buffer (pH 7.5). After centrifuging the extract for 30 min, the activity of γ-GK was measured according to Hayzer and Leisinger [100] in an assay mixture containing a 50 mM Tris buffer (pH 7.0), L-glutamate, MgCl_2_, ATP, hydroxamate–HCl and the enzyme. After terminating the reaction with a stop buffer containing FeCl_3_ and TCA, the absorbance was read at 535 nm.

### 4.4. Measurement of Oxidative Stress Parameters

Lipid peroxidation was measured following the method of Heath and Packer [101]. Fresh 100 mg leaf tissue was macerated in 1% trichloro acetic acid (TCA) and the extract was centrifuged at 10,000× *g*. The supernatant (1.0 mL) was reacted with 4 mL thiobarbituric acid for half an hour at 95 °C. After cooling the samples in an ice bath, centrifugation was carried out at 5000× *g* for 5 min and the absorbance was measured at 532 and 600 nm.

For the estimation of hydrogen peroxide content, 100 mg fresh tissue was extracted in 0.1% TCA and homogenate was centrifuged at 12,000× *g*. After mixing the supernatant with potassium phosphate buffer (pH 7.0) and potassium iodide, the optical density was taken at 390 nm [102]. The superoxide (O_2_^−^) was determined by extracting 100 mg fresh tissue in 65 mM potassium phosphate buffer (pH 7.8) and the absorbance was taken at 530 nm [103].

### 4.5. Assay of Protease and Lipoxygenase Activity

For assaying activity of protease (EC 3.4.21.40), the method of Green and Neurath [104] was used and fresh tissue was extracted in a chilled 50 mM sodium potassium buffer (pH 7.4) containing PVP. After centrifuging the homogenate at 5000× *g* for 5 min at 4 °C, the supernatant was incubated with casein at 40 °C and the amount of tyrosine released was determined by reaction with Folin–Ciocalteu’s reagent. The absorbance was taken at 660 nm. For measuring the activity of lipoxygenase (LOX; EC, 1.13.11.12), the method by Doderer et al. [105] was employed and the absorbance was taken at 234 nm using linoleic acid as a substrate, and the extinction coefficient of 25 mM^−1^ cm^−1^ was used to calculate.

### 4.6. Assay of Antioxidant Enzymes

For the extraction of antioxidant enzymes, fresh 1.0 gm leaf tissue was extracted in cold phosphate buffer (100 mM; pH 7.8) and supplemented with 1% PVP, 1 mM EDTA and 0.1 mM PMSF using a prechilled pestle and mortar. The homogenate was centrifuged at 12,000× *g* for 15 min at 4 °C and the supernatant was collected and used as the enzyme source.

The activity of superoxide dismutase (SOD, EC 1.15.1.1) was measured according to Bayer and Fridovich [106] and the photoinhibition of nitroblue tetrazolium by the enzyme was read at 560 nm after incubating the samples for 15 min under light.

For assaying the activity of ascorbate peroxidase (APX, EC 1.11.1.11), the method by Nakano and Asada [107] was followed and the absorbance was taken at 290 nm for 3 min.

The activity of glutathione reductase (GR; EC 1.6.4.2) was assayed according to Foyer and Halliwell [108] and the absorbance was taken at 340 nm for 2 min.

The activity of dehydroascorbate reductase (DHAR; EC 1.8.5.1) was measured according to Nakano and Asada [107] and the absorbance was taken at 265 nm for 2 min.

### 4.7. Estimation of Content of Ascorbate, Reduced Glutathione and Tocopherol

For the determination of ascorbic acid content, the method by Mukherjee and Choudhuri [109] was employed while the reduced glutathione (GSH) content was determined according to Ellman [110]. For calculation standards, AsA and GSH were used. Tocopherol was extracted in ethanol and petroleum ether (1.6:2) and the optical density was measured at 520 nm [111]. The standard curve of tocopherol was used for calculation.

### 4.8. Estimation of Nitric Oxide and Hydrogen Sulphide

Nitric oxide (NO) was determined according to Zhou et al. [112]. Briefly, 500 mg of fresh tissue was extracted in an ice-cold 50 mM acetic acid buffer (pH 3.6) supplemented by 4% zinc diacetate. The extract was centrifuged at 11,500× *g* for 15 min and the supernatant was neutralised by charcoal. After filtering the samples, 1 mL Greiss reagent was added to the filtrate and left at room temperature for 30 min, followed by measurement of optical density at 540 nm. The standard curve of sodium nitrite (NaNO_2_) was used for calculation.

For the estimation of hydrogen sulphide (HS), the method described by Nashef et al. [113] was followed. After extracting, 300 mg tissue in 100 mM potassium phosphate buffer (7.0) was used containing EDTA (10 mM). Homogenate was centrifuged at 15,000× *g* for 15 min and supernatant (100 μL) was mixed with 1880 μL extraction buffer and 20 μL of 5, 5-dithiobis (2-nitrobenzoicacid). After incubating the mixture at 25 °C for 5 min, the absorbance was taken at 412 nm and a standard curve of NaHS was used for calculation.

### 4.9. Measurement of Nitrate Reductase, Glutamine Synthetase and Glutamate Synthase Activity

The method described by Shrivastava [114] was used for assaying the activity of nitrate reductase (NR; EC 1.6.6.1) by incubating freshly cut leaf tissue in 100 mM potassium phosphate buffer (pH 7.5) containing 200 mM KNO_3_ and 0.5% n-propanol (*v*/*v*) for 3 h. The absorbance was taken at 540 nm. For measuring the activity of glutamine synthetase (GS; EC 6.3.1.2), the method described by Agbaria et al. [115] was followed and the formation of γ-glutamylhydroxamate was read at 540 nm. The activity of NADH-glutamate synthase (NADH-GOGAT; EC 1.4.1.14) was determined by the method by Lea et al. [116] and the absorbance was taken at 340 nm.

### 4.10. Estimation of Total Phenols, Flavonoids and Activity of Phenylalanine Ammonia Lyase

The method by Malick and Singh [117] was used for the estimation of total phenols in a dry powdered (500 mg) sample. After extraction in 80% ethanol, the supernatant was reacted with a 1N Folin–Ciocalteu reagent and 1 mL Na_2_CO_3_. The absorbance was taken at 650 nm and the standard curve of catechol was used for calculation.

For the estimation of flavonoid content, the method by Zhishen et al. [118] was followed. Briefly, after extracting 100 mg dry powdered sample in methanol, 1 mL supernatant was reacted with 5% NaNO_2_ and 10% AlCl_3_, followed by the addition of NaOH and distilled water. Absorbance was taken at 510 nm and catechin was used as the standard.

The activity of phenylalanine ammonia lyase (PAL) was measured according to Zucker [119] and optical density was taken at 290 nm.

### 4.11. Estimation of Ions

Nitrogen was estimated according to micro-Kjeldahl method [120] and P, K and S were determined using an atomic absorption spectrophotometer.

### 4.12. Total RNA Extraction and Gene Expression Analysis (qRT-PCR)

Leaf samples collected from plants subjected to each treatment were rapidly frozen in liquid nitrogen and stored at −80 °C until use. The total RNA was extracted using TRIzol reagent (Invitrogen, Carlsbad, CA, USA), with the quality of the extracted RNA being examined using RNAse-free 1% agarose gel and an Agilent 2100 Bioanalyzer (Agilent Technologies, Santa Clara, CA, USA). The quantity of extracted RNA was determined using a NanoDrop 2000 Spectrophotometer (Thermo Fisher Scientific, Wilmington, DE, USA). The mungbean reference genome was downloaded from Ensembl (https://plants.ensembl.org/info/website/ftp/index.html (accessed on 15 August 2023 ). The primers used for RT-qPCR are listed in Appendix A. The sequences of all primers were designed using Primer 5.0 software. Using a Primer Script RT Reagent Kit (Takara, Dalian, China), cDNA was prepared from 1 μL RNA in a total reaction volume of 20 μL. RT-qPCR was performed in a 10 μL reaction mixture consisting of 0.5 μL SYBR Green (Bioer Technology Co., Ltd., Hangzhou, China), 0.2 μL forward and reverse primers, 0.4 μL cDNA, and 8.9 μL water. PCR was carried out at 94 °C for 3 min, followed by 40 cycles of 94 °C for 10 s and 60 °C for 30 s, and finally, at 72 °C for 5 min. The mungbean *β*-Actin gene were used as an internal control in RT-PCR experiments [121]. Relative gene expression was calculated according to the 2^−ΔΔCt^ method [122]. The experiment was performed with three biological replicates, each with three technical replicates.

### 4.13. Statistical Analysis

The mean (±SE) value of three replicates is given and the least significant difference (LSD) among the mean values was determined at *p* < 0.05 using one-way ANOVA. Correlation analyses were performed using the R v.3.5.2 (Development Core Team R, 2016).

## 5. Conclusions

Conclusively, exogenous NO and the supplementation of S proved beneficial in alleviating the damaging effects of drought stress in *Vigna radiata*. Drought stress reduced chlorophylls synthesis, photosynthesis and the activity of N-metabolising enzymes. Treatments of NO and S up-regulated the tolerance mechanisms to alleviate the PEG-induced oxidative effects on membranes, photosynthesis and enzyme activity. The alleviation of oxidative damage was correlated with up-regulated antioxidant functioning, osmolyte and secondary metabolite accumulation. The mineral uptake and assimilation also benefited from NO and S treatments.

## Figures and Tables

**Figure 1 plants-12-03082-f001:**
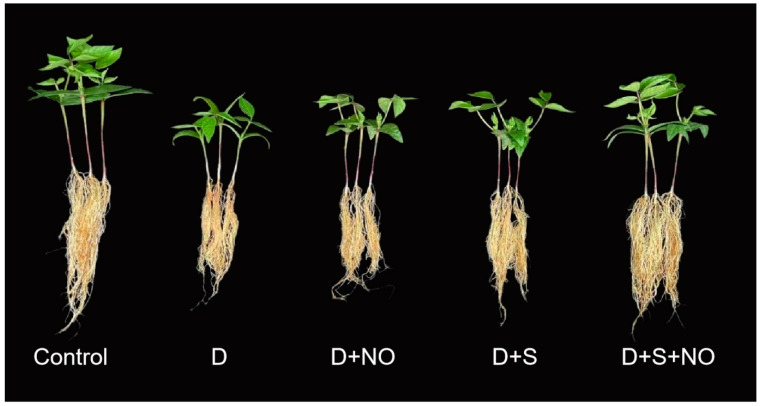
Photograph showing growth of mungbean treated with polyethylene glycol (D), sulphur (S) and nitric oxide (NO).

**Figure 2 plants-12-03082-f002:**
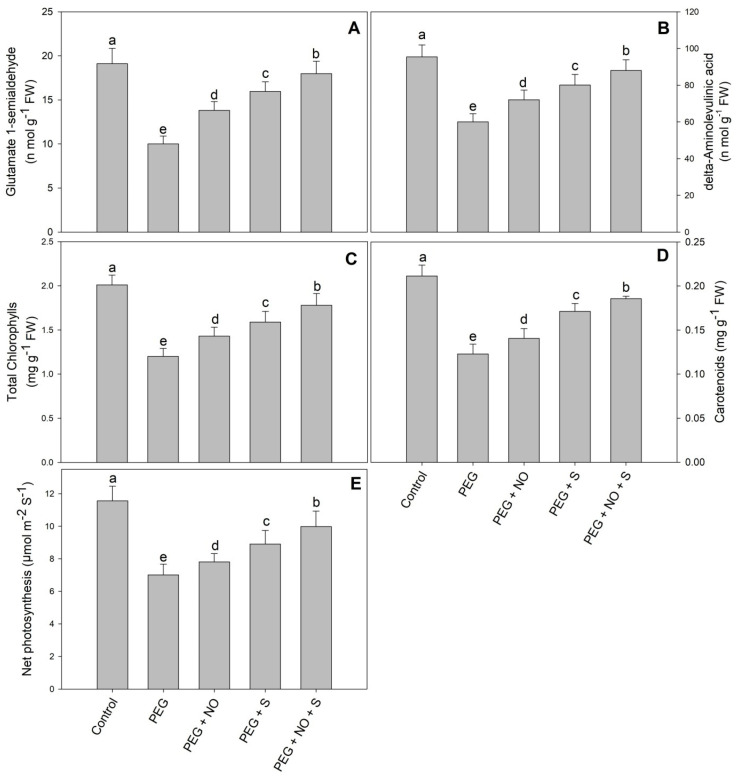
Effect of NO and sulphur (individual and combined) on the content of (**A**) glutamate 1-semialdehyde, (**B**) amino levulinic acid, (**C**) total chlorophyll, (**D**) carotenoids and (**E**) photosynthesis in *Vigna radiata* cultivar Jin 8 under PEG-induced drought stress. Data are mean (±SE) of three replicates and different letters show significant difference at *p* < 0.05.

**Figure 3 plants-12-03082-f003:**
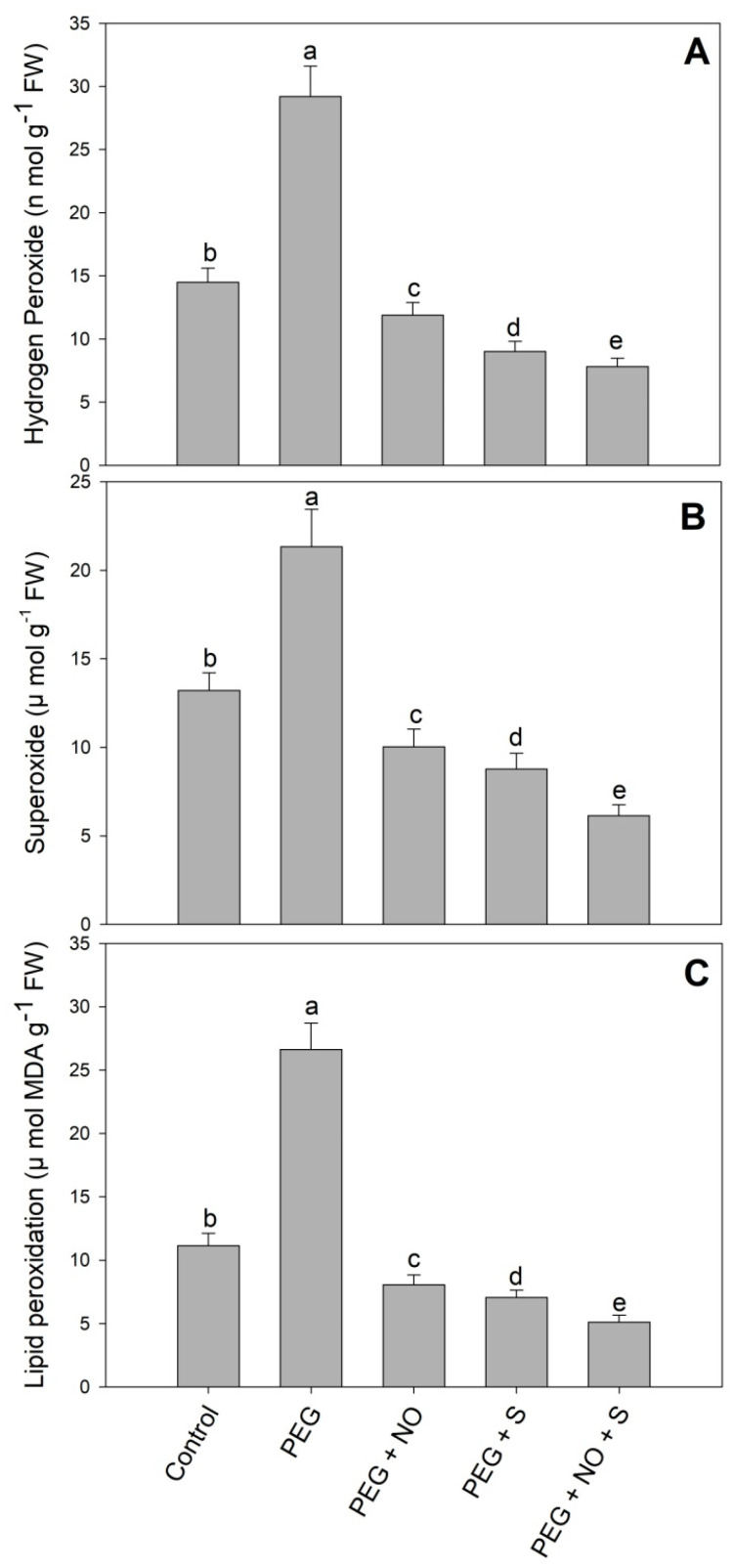
Effect of NO and sulphur (individual and combined) on the content of (**A**) hydrogen peroxide, (**B**) superoxide and (**C**) lipid peroxidation in *Vigna radiata* cultivar Jin 8 under PEG-induced drought stress. Data are mean (±SE) of three replicates and different letters show significant difference at *p* < 0.05.

**Figure 4 plants-12-03082-f004:**
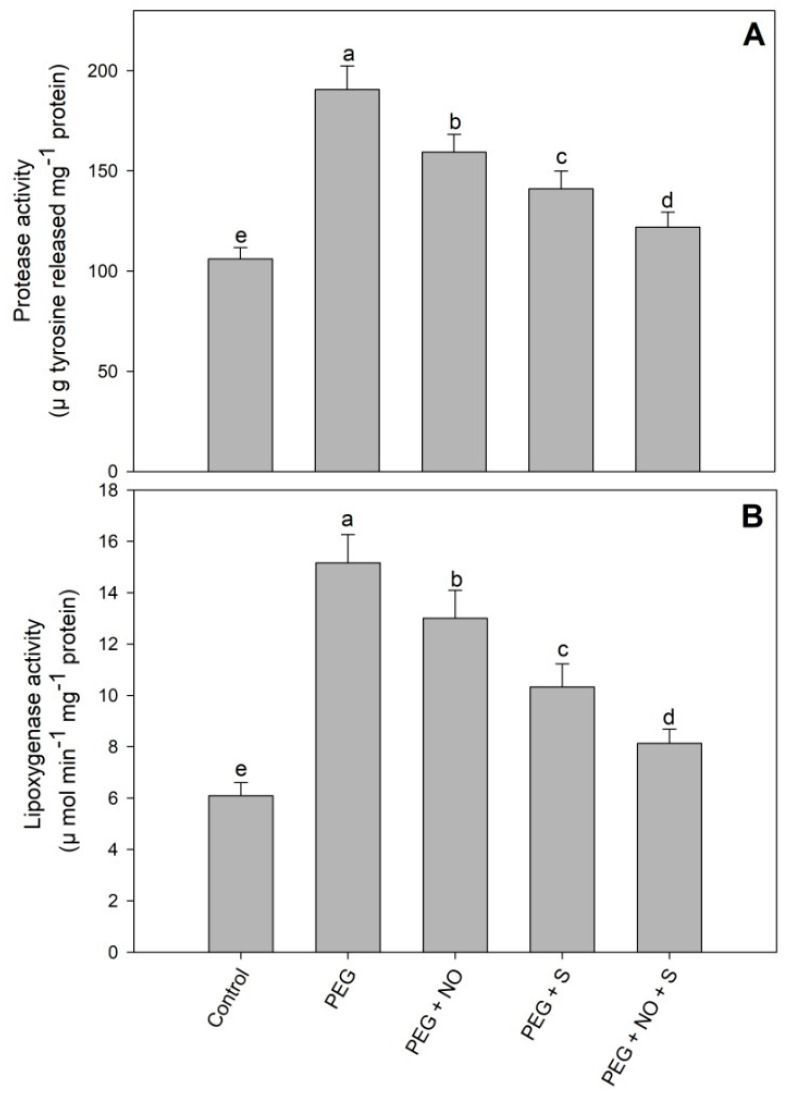
Effect of NO and S sulphur (individual and combined) on the activity of (**A**) protease and (**B**) lipoxygenase in *Vigna radiata* cultivar Jin 8 under PEG-induced drought stress. Data are mean (±SE) of three replicates and different letters show significant difference at *p* < 0.05.

**Figure 5 plants-12-03082-f005:**
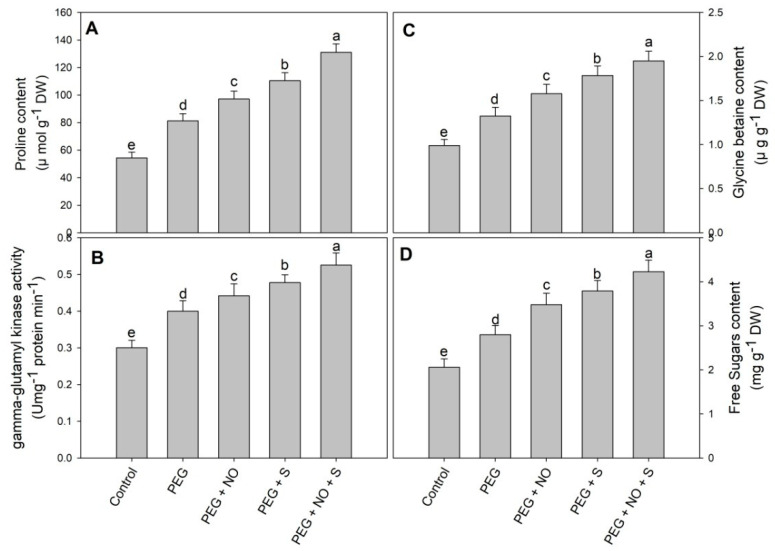
Effect of NO and sulphur(individual and combined) on (**A**) free proline, (**B**) activity of γ-glutamyl kinase, (**C**) glycine betaine and (**D**) sugars in *Vigna radiata* cultivar Jin 8 under PEG-induced drought stress. Data are mean (±SE) of three replicates and different letters show significant difference at *p* < 0.05.

**Figure 6 plants-12-03082-f006:**
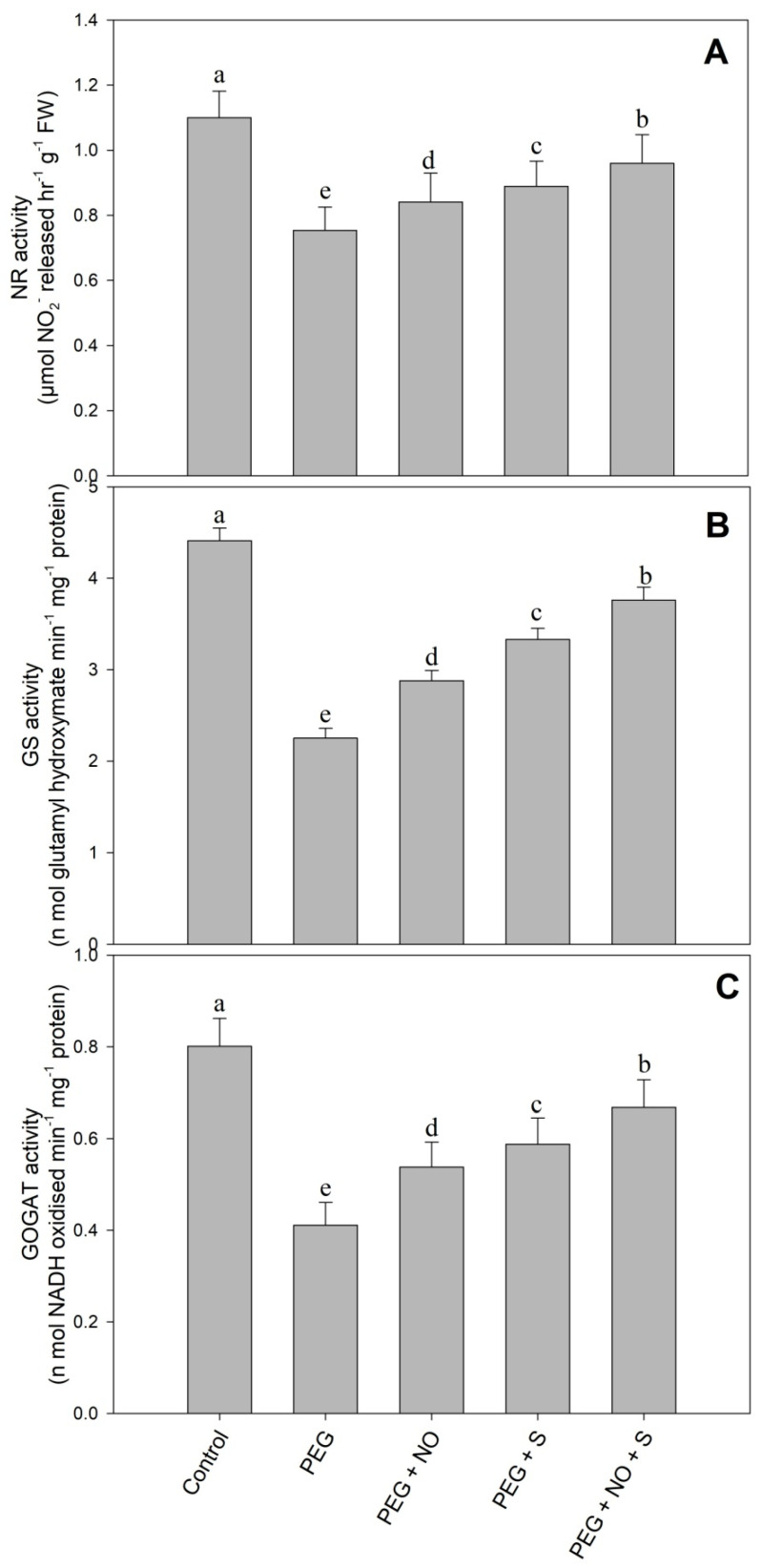
Effect of NO and sulphur (individual and combined) on activity of (**A**) nitrate reductase, (**B**) glutamate synthase and (**C**) glutamine synthetase in *Vigna radiata* cultivar Jin 8 under PEG-induced drought stress. Data are mean (±SE) of three replicates and different letters show significant difference at *p* < 0.05.

**Figure 7 plants-12-03082-f007:**
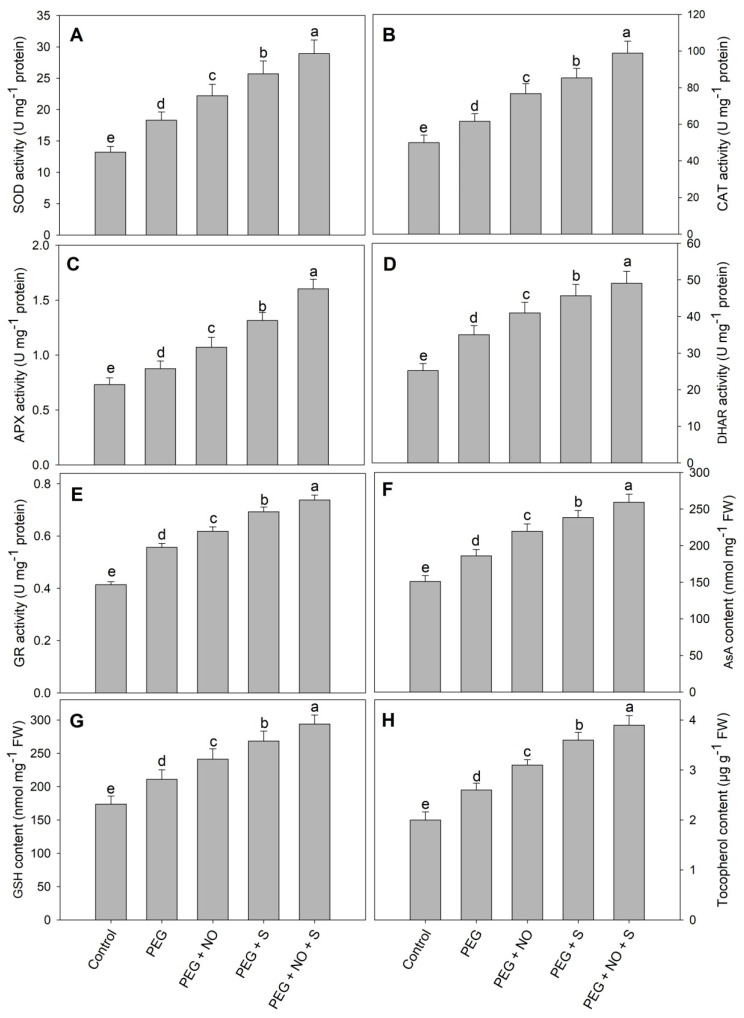
Effect of NO and sulphur (individual and combined) on activity of (**A**) superoxide dismutase, (**B**) catalase, (**C**) ascorbate peroxidase, (**D**) monodehydroascorbate reductase, (**E**) glutathione reductase and (**F**) content of ascorbic acid, (**G**) reduced glutathione and (**H**) tocopherol in *Vigna radiata* cultivar Jin 8under PEG-induced drought stress. Data are mean (±SE) of three replicates and different letters show significant difference at *p* < 0.05.

**Figure 8 plants-12-03082-f008:**
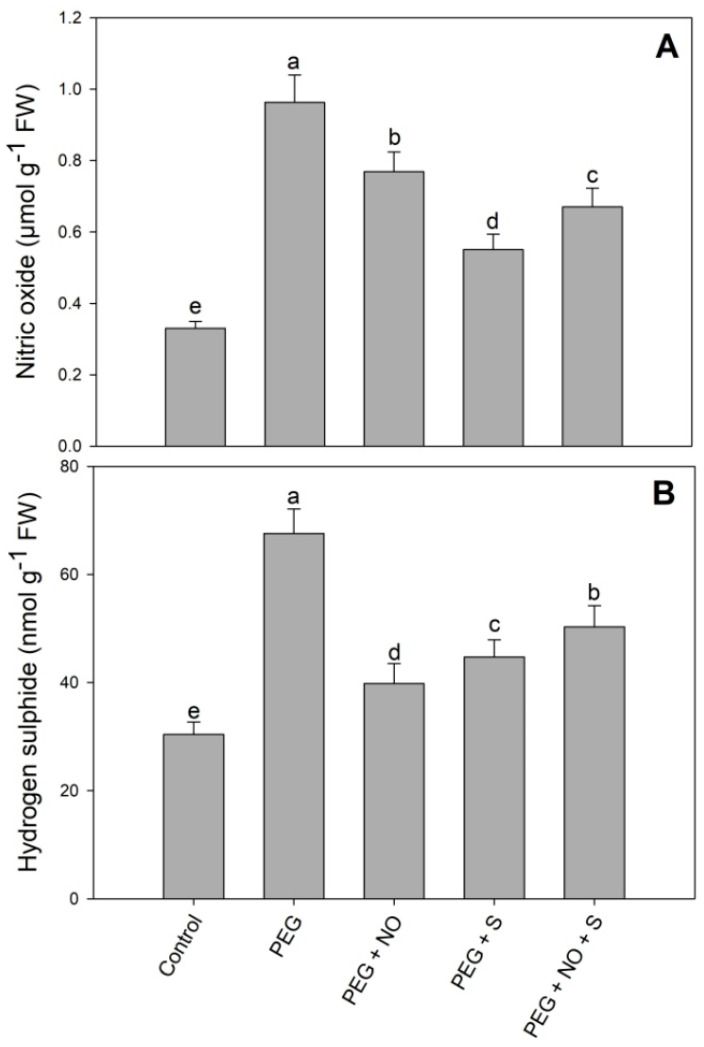
Effect of NO and sulphur (individual and combined) on endogenous concentration of (**A**) nitric oxide and (**B**) hydrogen sulphide in *Vigna radiata* cultivar Jin 8under PEG-induced drought stress. Data are mean (±SE) of three replicates and different letters show significant difference at *p* < 0.05.

**Figure 9 plants-12-03082-f009:**
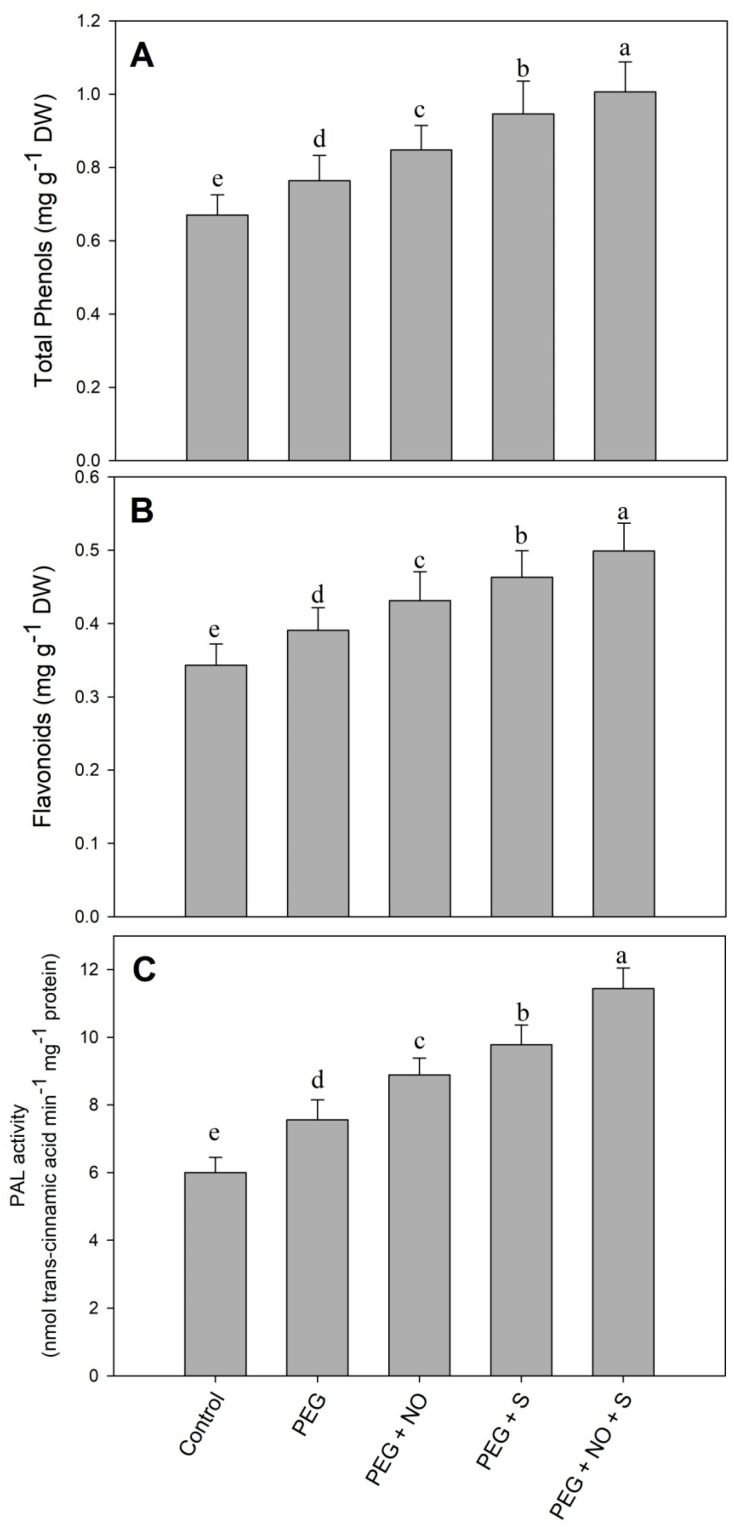
Effect of NO and sulphur (individual and combined) on content of (**A**) total phenols, (**B**) flavonoids and activity of (**C**) phenylalanine ammonia lyase in *Vigna radiata* cultivar Jin 8 under PEG-induced drought stress. Data are mean (±SE) of three replicates and different letters show significant difference at *p* < 0.05.

**Figure 10 plants-12-03082-f010:**
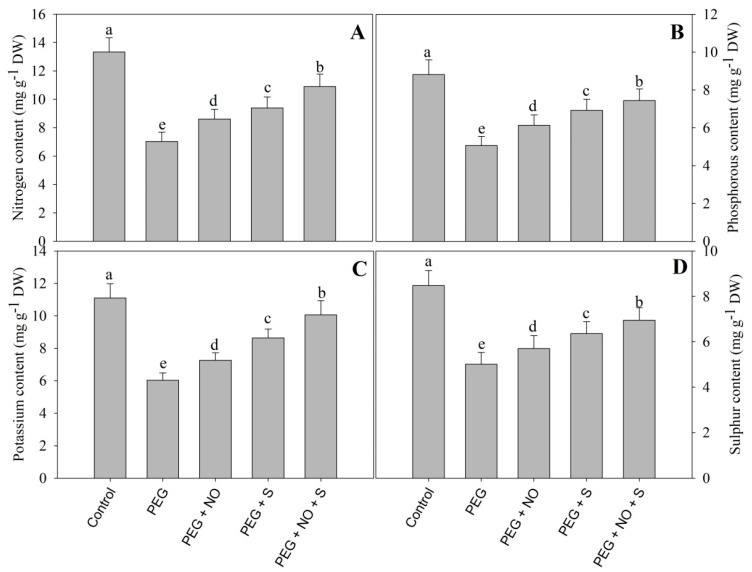
Effect of NO and sulphur (individual and combined) on content of (**A**) nitrogen, (**B**) phosphorous, (**C**) potassium and (**D**) sulphur in *Vigna radiata* cultivar Jin 8 under PEG-induced drought stress. Data are mean (±SE) of three replicates and different letters show significant difference at *p* < 0.05.

**Figure 11 plants-12-03082-f011:**
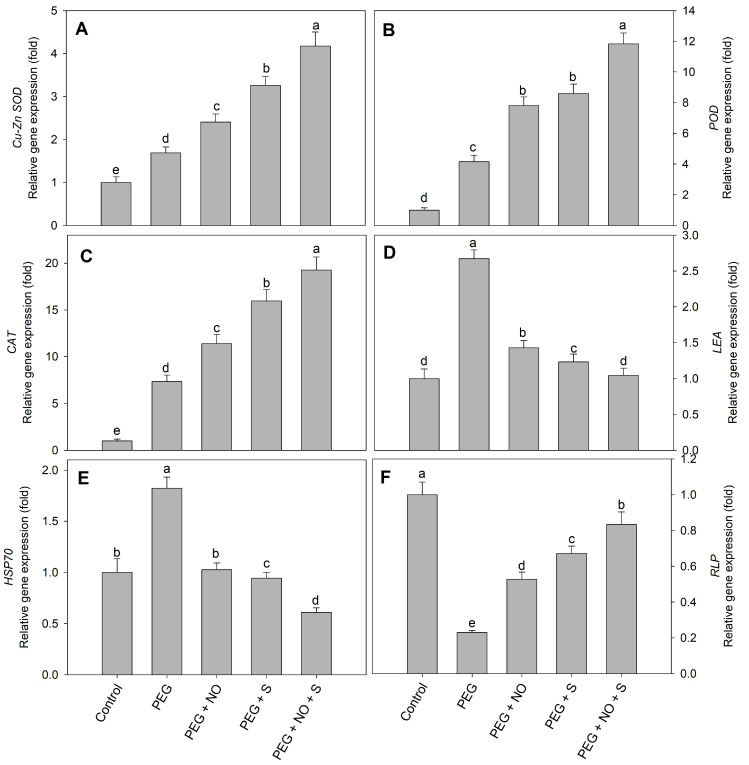
Effect of NO and sulphur (individual and combined) on content of gene expression of (**A**) *Cu/ZnSOD*, (**B**) *POD*, (**C**) *CAT*, (**D**) *LEA*, (**E**) *HSP70* and (**F**) *RLP* in *Vigna radiata* cultivar Jin 8 under PEG-induced drought stress. Data are mean (±SE) of three replicates and different letters show significant difference at *p* < 0.05.

**Figure 12 plants-12-03082-f012:**
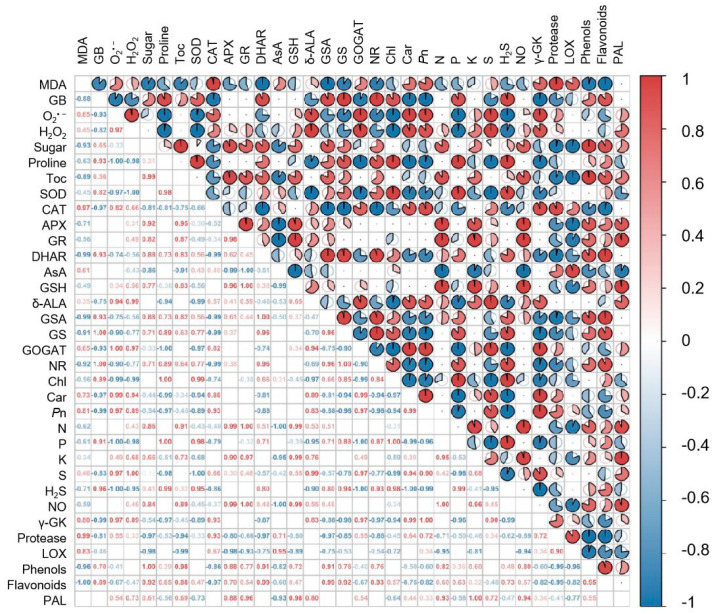
Correlation analysis between physiological parameters of mungbean after S and NO application under drought stress. Abbreviations: MDA, malonaldehyde; GB, glycine betaine; O_2_^−^, superoxide; H_2_O_2_, hydrogen peroxide; Toc, tocopherol; SOD, superoxide dismutase; CAT, catalase; APX, ascorbate peroxidase; GR, glutathione reductase, DHAR, dehydroascorbate reductase; AsA, ascorbic acid; GSH, reduced glutathione; δ-ALA, δ-aminolevulinic acid; GSA, glutamate 1-semialdehyde; GS, glutamine synthetase; GOGAT, glutamate synthase; NR, nitrate reductase; Chl, total chlorophylls; Car, carotenoids; Pn, photosynthesis; N, nitrogen, P, phosphorous; K potassium; S, sulphur; H_2_S, hydrogen sulphide; NO, nitric oxide; γ-GK, γ-glutamyl kinase; LOX, lipoxygenase; PAL, phenylalanine ammonia lyase.

## Data Availability

All data generated have been include in manuscript.

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
