# Peer review of "Alleviation of Adverse Effects of Drought Stress on Growth and Nitrogen Metabolism in Mungbean (Vigna radiata) by Sulphur and Nitric Oxide Involves Up-Regulation of Antioxidant and Osmolyte Metabolism and Gene Expression"

_plants, 2023, doi:10.3390/plants12173082_

Round 1

Reviewer 1 Report

Introduction

Reduce the first part about general and physiological effects of drought and increase the second part mention any transcriptomic or gene expression studies in legumes in response to drought along with sulphur and nitric oxide. End of the introduction, write the objectives clearly. 

Materials and methods

Write the cultivar name and source of the seeds, pot size, and how many plants per pot. Explain how it was chosen Actin as the housekeeping gene. Has the calibration curve to detect any aspecific amplification performed? Have any primers for real-time tested using a calibration curve based on the amount of DNA used? The three independent replicates are biological, or not? How many technical replicates have been used? Would be possible to have the raw Ct data to understand the variability between technical replicates?

Motivate the choice of the targeted genes, especially those encoding transcription factors. They are a family of genes, why these members were chosen instead of the other members.

How is possible to be sure that these primers amplify only these members and not the others of the same family? How primers have been designed?

Authors MUST provide full information about all used primers. This information have to include: (A) Names of primers; (B) Sequence accession numbers from the database, for example, NCBI (like: AB12345) or Legumeinfo (like: Peroxidase = vigna.Vradi0007s00360), or another database for each gene studied; (C) Sequences of both F and R primers; (D) Amplicon sizes for each used qPCR product; (E) Reference gene Actin must be also included in this list with all required information; (F) If any of the presented information about primers and qPCR in mung-bean (including reference gene Actin) was published, please provide references. This information has to be present in the Supplementary file with full details.

Results and discussion

This section needs subsequent improvement. Try to integrate more the physiological and biochemical data with the gene expression data providing other previous data on similar crops, eventually legumes.

Editing of english language required

Author Response

Introduction

Reduce the first part about general and physiological effects of drought and increase the second part mention any transcriptomic or gene expression studies in legumes in response to drought along with sulphur and nitric oxide. End of the introduction, write the objectives clearly.

As suggested few sentences about transcriptomic studies about NO and S were added. However, we request please to allow us maintain the introduction as such to keep flow of physiology and biochemistry part consistant.

Materials and methods

Write the cultivar name and source of the seeds, pot size, and how many plants per pot.

As suggested the details have been given in revised paper.

Explain how it was chosen Actin as the housekeeping gene. Has the calibration curve to detect any aspecific amplification performed? Have any primers for real-time tested using a calibration curve based on the amount of DNA used? The three independent replicates are biological, or not? How many technical replicates have been used? Would be possible to have the raw Ct data to understand the variability between technical replicates?

The details about primers have been given. We used Actin as house keeping from previous read papers ((Yu et al. 2022 and Ma et al. 2023).

YuML , Huang L, Feng NJ, ZhengDF, Zhao JJ. 2022. Exogenous Uniconazole Enhances Tolerance to Chilling Stress In Mung Beans (Vigna Radiata L.) Through Cross Talk Among Photosynthesis, Antioxidant System, Sucrose Metabolism, and Hormones. J plant physiol. 276, 153772. doi: 10.1016/j.jplph.2022.153772).

Ma C,FengYL ,Zhou S,Zhang J, GuoBB ,Xiong Y, WuSW,Li Y, LiYJ , LiCX ,Metabolomics and transcriptomics provide insights into the molecular mechanisms of anthocyanin accumulation in the seed coat of differently colored mung bean (Vigna radiata L.),2023,Plant PhysiolBiochem,200,107739,https://doi.org/10.1016/j.plaphy.2023.107739)

The efficiency of the primers for gene expression were calculated. According to the reaction results of PCR, draw a standard curve. 10-fold gradient dilution of each primer (divided into 100, 10-1, 10-2, 10-3, 10-4, 10-5 concentration gradients) was performed, and quantitative PCR was performed to generate a gene expression standard curve. The abscissa of the standard curve is the log value of the initial concentration of the template, the ordinate is the Ct value, the formula E= 10-1/slope-1)×100% represents the amplification efficiency, and the Slope is the slope of the standard curve. After RT-PCR amplification, it is found that the designed primers can amplify to specific fragments, and the melting curve graph is generated at 60℃-90℃. The specificity is good, which meets the requirements of the follow-up test. The linear reliability R2 of the standard curve of each gene is higher than the default value of 0.998-1.000,so the obtained amplification efficiency E range is in line with 93.1-102.6%, thus ensuring the efficiency of amplification, and the result is in line with real-time fluorescence quantification scope of use.

The formula used for the expression of the genes in our study is as follows

Ct targetCt reference = ΔCt

ΔCt treatmentΔCt control = ΔΔCt

Relative value (Fold change) = 2-ΔΔCt

Motivate the choice of the targeted genes, especially those encoding transcription factors. They are a family of genes, why these members were chosen instead of the other members.

Response: In these gene families, the selected genes were specific.

How is possible to be sure that these primers amplify only these members and not the others of the same family? How primers have been designed?

The sequences of all primers were designed using Primer 5.0 software. Primers were evaluated using the Primer-blast in NCBI. Then primer specificity analysis was performed through the RT-PCR amplification.

Authors MUST provide full information about all used primers. This information have to include: (A) Names of primers; (B) Sequence accession numbers from the database, for example, NCBI (like: AB12345) or Legume info (like: Peroxidase = vigna.Vradi0007s00360), or another database for each gene studied; (C) Sequences of both F and R primers; (D) Amplicon sizes for each used qPCR product; (E) Reference gene Actin must be also included in this list with all required information; (F) If any of the presented information about primers and qPCR in mung-bean (including reference gene Actin) was published, please provide references. This information has to be present in the Supplementary file with full details.

The details have been provided in revised supplementary file (Table S1)

Results and discussion

This section needs subsequent improvement. Try to integrate more the physiological and biochemical data with the gene expression data providing other

As suggested by worthy reviewer the section has been revised thoroughly.

Reviewer 2 Report

The authors have investigated the alleviation of drought stress by nitric oxide and sulphur in Vigna radiata. The main results included that application of nitric oxide and sulphur alleviated the decline in morphological characters in seedlings; reduction in intermediates of chlorophyll synthesis pathways and photosynthesis; PEG induced oxidative stress was assuaged; reduction in the activity of nitrate reductase, glutamine synthetase and glutamate synthase; PEG induced increase in NO and HS was lowered; enhanced antioxidant system, osmolyte and secondary metabolite accumulation; activity of γ-glutamyl kinase and phenylalanine ammonia lyase was up-regulated; significantly regulated gene expression of Cu/ZnSOD, POD, CAT, RBCL, HSP70 and LEA. The authors concluded with the beneficial use of NO and S in mitigation of drought induced alterations in metabolism of Vigna radiata.

I believe that the overall presentation is fine, though a comprehensive editorial revision is necessary to correct the substantial amounts of editorial errors throughout the entire manuscript. More detailed suggestions are listed below for the authors to consider if a revision is requested by the editor.

Title: please provide the latin name of mungbean, Vigna radiata, in the title

Abstract: I would like to recommend that the descriptions of the effects of PEG be shortened.

Line 23: “interms”?

Line 27: “was increased”

Line 30: activities

Throughout the entire manuscript, I would like to keep the full spelling of “sulphur” instead of the abbreviation “S”

Introduction: I believe that the authors provided sufficient background

Results: The figures are presented well. However, I believe that Plate 1 should be presented in Results.

Figure 1:

Line 104, there is an extra space that needs to be deleted.

Line 104: Data are…

Line 105: P < 0.05 (extra spaces are needed)

These above problems are also shown in many other figures…

Discussion: The authors have concluded appropriately based on available data.

I would suggest that the authors establish a few subsections in Discussion to focus on the in-depth discussion of these subjects.

Materials and Methods:

Line 579: “qRT-PCR” is out of place?

I believe that the overall presentation is fine, though a comprehensive editorial revision is necessary to correct the substantial amounts of editorial errors throughout the entire manuscript.

Author Response

The authors have investigated the alleviation of drought stress by nitric oxide and sulphur in Vigna radiata. The main results included that application of nitric oxide and sulphur alleviated the decline in morphological characters in seedlings; reduction in intermediates of chlorophyll synthesis pathways and photosynthesis; PEG induced oxidative stress was assuaged; reduction in the activity of nitrate reductase, glutamine synthetase and glutamate synthase; PEG induced increase in NO and HS was lowered; enhanced antioxidant system, osmolyte and secondary metabolite accumulation; activity of γ-glutamyl kinase and phenylalanine ammonia lyase was up-regulated; significantly regulated gene expression of Cu/ZnSOD, POD, CAT, RBCL, HSP70 and LEA. The authors concluded with the beneficial use of NO and S in mitigation of drought induced alterations in metabolism of Vigna radiata.

I believe that the overall presentation is fine, though a comprehensive editorial revision is necessary to correct the substantial amounts of editorial errors throughout the entire manuscript. More detailed suggestions are listed below for the authors to consider if a revision is requested by the editor.

Title: please provide the latin name of mungbean, Vigna radiata, in the title

Has been done

Abstract: I would like to recommend that the descriptions of the effects of PEG be shortened.

Has been shortened as suggested.

Line 23: “interms”?

Has been deleted and sentence modified.

Line 27: “was increased”

Has been modified as suggested

Line 30: activities

Has been modifies as suggested

Throughout the entire manuscript, I would like to keep the full spelling of “sulphur” instead of the abbreviation “S”

Has been done as suggested

Introduction: I believe that the authors provided sufficient background

Thanks

Results: The figures are presented well. However, I believe that Plate 1 should be presented in Results.

Thanks, as suggested plate I in results.

Figure 1:

Line 104, there is an extra space that needs to be deleted.

Deleted as suggested

Line 104: Data are…

Modified as suggested

Line 105: P < 0.05 (extra spaces are needed)

Modified as suggested

These above problems are also shown in many other figures…

Corrected in all.

Discussion: The authors have concluded appropriately based on available data.

I would suggest that the authors establish a few sub

Thanks for the encouragement

Round 2

Reviewer 1 Report

I recommend the manuscript for publication with language correction

Minor editing of English language required

Reviewer 2 Report

I appreciate the efforts that the authors have devoted to improving their manuscript. I still see some editorial errors and the authors did not respond to my comments for a comprehensive proofreading of their manuscript to correct the editorial errors. I guess these editorial errors could be corrected during the proofreading stage if the manuscript is accepted for publication. I have no more questions.

This has been indicated in my review for the authors.